# ON-POLICY RL WITH OPTIMAL REWARD BASELINE

## ABSTRACT

Reinforcement learning algorithms are fundamental to align large language models with human preferences and to enhance their reasoning capabilities. However, current reinforcement learning algorithms often suffer from training instability due to loose on-policy constraints and computational inefficiency due to auxiliary models. In this work, we propose *On-Policy RL with Optimal reward baseline* (**OPO**), a novel and simplified reinforcement learning algorithm designed to address these challenges. OPO emphasizes the importance of exact on-policy training, which empirically stabilizes the training process and enhances exploration. Moreover, OPO integrates a practically feasible formulation of the optimal reward baseline that minimizes gradient variance. We evaluate OPO on mathematical reasoning benchmarks. The results demonstrate its superior performance and training stability without additional models or regularization terms. Furthermore, OPO achieves **lower policy shifts** and **higher output entropy**, encouraging **more diverse and less repetitive responses**. These results highlight OPO as a promising direction for stable and effective reinforcement learning in large language model alignment and reasoning tasks. The OPO implementation is integrated into the VeRL library.

## 1 INTRODUCTION

Reinforcement learning from human feedback (RLHF) is a foundational approach for aligning large language models (LLMs) with human preferences (Stiennon et al., 2020; Ouyang et al., 2022; Bai et al., 2022). The standard RLHF pipeline typically involves supervised fine-tuning followed by reinforcement learning, commonly employing proximal policy optimization (PPO) algorithm (Schulman et al., 2017), guided by a learned reward model. Beyond general alignment, reinforcement learning has proven effective in enhancing the reasoning abilities of LLMs through test-time scaling, as demonstrated by the OpenAI-o1 model (OpenAI, 2024). Most recent work such as DeepSeek-R1 (Guo et al., 2025) further shows that reinforcement learning, even with simple rule-based rewards, can elicit emergent reasoning behaviors and significantly boost performance on complex tasks like mathematics and code generation.

Despite its success, current RLHF algorithms face some challenges regarding stability and efficiency. For instance, PPO (Schulman et al., 2017) requires training an extra value model to estimate advantages, which introduces additional computational overhead. While methods like Group Relative Policy Optimization (GRPO) address this by using response groups to compute a relative reward baseline (Shao et al., 2024), these methods are often prone to instability due to loose on-policy constraints. This often results in large policy shifts and reduced sample diversity, a phenomenon known as alignment tax (Askell et al., 2021; Kirk et al., 2024).

In this work, we introduce *On-Policy RL with Optimal reward baseline* (**OPO**), a simple yet effective algorithm with two key improvements. First, OPO employs exact on-policy training, which empirically stabilizes the training process and significantly enhances exploration capabilities. Second, we incorporate the optimal reward baseline that theoretically minimizes gradient variance. While the original optimal baseline is impractical, we derive a simplified form under intuitive assumptions, which makes it feasible for practical use. By integrating these improvements, OPO eliminates the need for auxiliary components such as value and reference models, as well as regularization terms. Instead, it relies solely on a single policy model optimized directly to maximize the expected reward.

We validate the effectiveness of OPO on Deepseek-R1-Distill-Qwen-7B model across various mathematical reasoning benchmarks. Our experimental results demonstrate that OPO outperforms existing baselines in both performance and training stability. In particular, OPO consistently maintains lower policy shifts and higher output entropy, leading to more diverse and less repetitive responses. In summary, the key advantages of OPO are:

- **Theoretical Soundness:** We incorporate the optimal reward baseline for sequence generation problems, which theoretically minimizes the gradient variance and practically easy to implement.

- **Enhanced Stability:** OPO exhibits stable training dynamics, even without explicit KL or entropy regularization, which is crucial for reliable performance.

- **Empirical Effectiveness:** OPO achieves better performance on math reasoning benchmarks and yields more diverse and less repetitive responses.

## 2 BACKGROUND

**Proximal Policy Optimization (PPO)**   PPO (Schulman et al., 2017) is a widely adopted policy gradient algorithm. As an actor-critic method, PPO leverages a policy model (actor) to optimize the reward and a value model (critic) to estimate the value of each state. A central feature of PPO is its clipped surrogate objective function, designed to enhance training stability and sample efficiency by limiting the magnitude of policy updates at each iteration. The objective is formally defined as:

$$
\mathcal{J}_{\text{PPO}}(\theta) = \mathbb{E}_{x \sim \mathcal{D}, y \sim \pi_\theta(\cdot|x)} \left[ \sum_{t=1}^{|y|} \left\{ \min(w_t \cdot A_t, \text{clip}(w_t, 1-\epsilon, 1+\epsilon) \cdot A_t) \right\} \right]
$$

$$
w_t = \frac{\pi_\theta(y_t|x, y_{<t})}{\pi_{\theta_{\text{old}}}(y_t|x, y_{<t})}
$$

(1)

where $w_t$ is the importance ratio, and $A_t$ denotes the advantage estimate at time step $t$, computed using Generalized Advantage Estimation (GAE, Schulman et al. 2018), which combines information from the reward function and the value function. The hyperparameter $\epsilon$ controls the clipping range, effectively constraining the policy update to prevent drastic changes that can make training unstable.

**Group Relative Policy Optimization (GRPO)**   To eliminate the computational cost of a separate value model, GRPO (Shao et al., 2024) computes relative advantages within a group of sampled responses by normalizing rewards. For each input $x$, GRPO samples a group of $K$ trajectories $\{y_i\}_{i=1}^K$ from the policy and defines the advantage of each trajectory based on its reward relative to others in the group:

$$
\hat{A}_{i,t} = \frac{r(x, y_i) - \text{mean}(\{r(x, y_i)\}_{i=1}^K)}{\text{std}(\{r(x, y_i)\}_{i=1}^K)}
$$

(2)

This group-wise normalization ensures zero mean and unit variance of advantages within each group, which achieves efficient training without requiring the additional value model. It extends the PPO objective with the relative advantage:

$$
\mathcal{J}_{\text{GRPO}}(\theta) = \mathbb{E}_{x \sim \mathcal{D}, \{y_i\}_{i=1}^K \sim \pi_{\theta_{\text{old}}}(\cdot|x)} \left[ \frac{1}{K} \sum_{i=1}^K \frac{1}{|y_i|} \sum_{t=1}^{|y_i|} \min\left( w_{i,t} \hat{A}_{i,t}, \text{clip}(w_{i,t}, 1-\epsilon, 1+\epsilon) \hat{A}_{i,t} \right) \right]
$$

$$
w_{i,t} = \frac{\pi_\theta(y_{i,t}|x, y_{i,<t})}{\pi_{\theta_{\text{old}}}(y_{i,t}|x, y_{i,<t})}
$$

(3)

**KL and Entropy Regularization**   In the reinforcement learning stage of RLHF, two regularization terms are commonly incorporated into the objective function to stabilize policy optimization: the Kullback-Leibler (KL) divergence loss and the entropy bonus. The KL divergence loss constrains the updated policy from drifting too far from a reference policy (typically the original supervised fine-tuned model) (Schulman, 2020). This constraint helps mitigate the alignment tax, which refers

to the degradation of helpfulness, safety, or factuality when the model over-optimizes for reward at the cost of its original capabilities.

In addition to the KL divergence loss, an entropy bonus is introduced to encourage exploration and prevent the policy to collapsing into a suboptimal solution (Ahmed et al., 2019). By maximizing the entropy of the policy distribution, we encourage the model to explore a broader set of potential high-reward responses, thus enhancing the diversity and robustness of the generated outputs.

Balancing these components (the primary reward objective, KL loss, and entropy bonus) is essential for achieving stable learning and maintaining both original capabilities and alignment performance. Over-penalizing with KL and entropy can limit learning progress, whereas under-penalizing can lead to undesirable policy drift. Similarly, entropy must be tuned to avoid both under-exploration and excessive randomness.

## 3 METHOD: ON-POLICY RL WITH OPTIMAL REWARD BASELINE (OPO)

We propose *On-Policy RL with Optimal reward baseline* (**OPO**), which employs two key strategies: (1) *exact on-policy training*, which we argue is crucial for mitigating issues like entropy collapse and large policy shifts in off-policy settings, and (2) the *optimal reward baseline* that theoretically minimizes gradient variance. OPO solely optimizes a policy model the maximize the expected reward without other regularization terms, which not only simplifies the training process but also leads to more stable and effective training compared to methods with loose on-policy settings and suboptimal baselines.

### 3.1 EXACT ON-POLICY TRAINING

The objective of policy-based reinforcement learning is to optimize a parameterized policy $\pi_\theta$ to maximize the expected reward. This objective is inherently on-policy, meaning that the reward expectation is taken with respect to trajectories generated directly by the current policy. Specifically, we aim to:

$$\max_\theta \mathbb{E}_{x \sim \mathcal{D}, y \sim \pi_\theta(\cdot|x)}[r(x,y)] \tag{4}$$

where $x$ is the input sampled from the dataset $\mathcal{D}$, $y$ represents a trajectory sampled from the current policy $\pi_\theta(\cdot|x)$, and $r(x,y)$ is the reward function for trajectory $y$ given input $x$. For simplicity, we mainly consider settings where the reward is trajectory-level.

A foundational characteristic of OPO is its strict adherence to exact on-policy training. This contrasts with common policy gradient methods, such as PPO, which typically collect a batch of data using the current policy and then perform multiple gradient updates on this fixed batch. While reusing rollouts can improve sample efficiency, subsequent updates introduce an off-policy divergence. In practice, it may contribute to sample entropy collapse and large policy shifts, thereby necessitating explicit entropy regularization. In contrast, exact on-policy training ensures that each gradient step is computed using fresh data sampled from the current policy. This preserves the theoretical properties of the policy objective and empirically leads to more stable entropy throughout training. Furthermore, exact on-policy training maintains a lower KL divergence between the current policy and the initial policy, reducing the alignment tax and improving the overall performance of the model.

### 3.2 LENGTH-WEIGHTED OPTIMAL REWARD BASELINE FOR VARIANCE REDUCTION

Reducing the variance of policy gradient estimates is crucial for stable and efficient reinforcement learning (Dayan, 1991; Weaver & Tao, 2001; Kakade & Langford, 2002; Greensmith et al., 2004). A common technique to reduce the variance is to subtract a baseline $b$ from the reward. Recall the policy gradient $g$ derived from the policy gradient theorem:

$$g = \mathbb{E}_{x \sim \mathcal{D}, y \sim \pi_\theta(\cdot|x)}[\nabla_\theta \log \pi_\theta(y|x) \cdot r(x,y)] \tag{5}$$

where $\nabla_\theta \log \pi_\theta(y|x)$ is the score function gradient. We can modify this gradient estimator to include a baseline $b$, which does not change the expected value of the gradient but can significantly reduce its variance. The modified gradient estimator becomes:

$$g = \mathbb{E}_{x \sim \mathcal{D}, y \sim \pi_\theta(\cdot|x)}[\nabla_\theta \log \pi_\theta(y|x) \cdot (r(x,y) - b)] \tag{6}$$

There exists the theoretical optimal baseline which can minimize the gradient variance. The variance is defined as:

$$\text{Var}[g] = \mathbb{E}[(\nabla_\theta \log \pi_\theta(y|x) \cdot (r(x, y) - b))^2] - (\mathbb{E}[\nabla_\theta \log \pi_\theta(y|x) \cdot (r(x, y) - b)])^2 \quad (7)$$

Since the second term (the square of the expected gradient) is independent of $b$, minimizing $\text{Var}[g]$ is equivalent to minimizing the first term. We can derive the optimal baseline $b^*$ by taking the derivative with respect to $b$ and setting it to zero:

$$\frac{d}{db}\mathbb{E}[(\nabla_\theta \log \pi_\theta(y|x) \cdot (r(x, y) - b))^2] = 0 \quad (8)$$

Solving this equation yields the optimal baseline $b^*$:

$$b^* = \frac{\mathbb{E}_{y\sim\pi_\theta(\cdot|x)}\left[(\nabla_\theta \log \pi_\theta(y|x))^2 \cdot r(x, y)\right]}{\mathbb{E}_{y\sim\pi_\theta(\cdot|x)}\left[(\nabla_\theta \log \pi_\theta(y|x))^2\right]} \quad (9)$$

This optimal baseline represents a weighted average of rewards, where the weights are the squared magnitudes of the score function gradients. This specific weighting minimizes the variance of our policy gradient estimate. The detailed derivation is provided in Appendix A. The computation of Equation 9 is impractical because it requires individual gradient norm calculations for each trajectory. Nevertheless, we demonstrate how to simplify this equation for practical sequence generation under a straightforward assumption.

**Practical Optimal Baseline for Sequence Generation**  For sequence generation problems, such as language modeling, we can make a simple assumption that the gradients of different tokens are approximately orthogonal and the norm of the gradient for each token follows a same distribution. Under this condition, the squared magnitude of the policy gradient for a trajectory is proportional to its length ($\|\nabla_\theta \log \pi_\theta(y|x)\|^2 \propto l_y$), where $l_y$ is the length of the response $y$. With this simplification, the optimal reward baseline simplifies to:

$$b^* = \frac{\mathbb{E}_{y\sim\pi_\theta(\cdot|x)}[l_y \cdot r(x, y)]}{\mathbb{E}_{y\sim\pi_\theta(\cdot|x)}[l_y]} \quad (10)$$

where longer responses contribute proportionally more to the baseline calculation. This formulation results in a length-weighted average of the reward, making it both theoretically sound and straightforward to compute in practice, which facilitates its integration into sequence generation problems with trajectory-level rewards.

## 3.3 Overall Algorithm

The OPO algorithm integrates the two key techniques discussed: exact on-policy training and the optimal reward baseline. In practice, we follow the GRPO setup: for each prompt, we sample $K$ outputs using the current policy, compute an approximation of the optimal baseline using these samples, and then perform policy optimization with exact on-policy training. Specifically, given a prompt $x$ and $K$ sampled responses $\{y_i\}_{i=1}^K$, the objective function of OPO can be expressed as:

$$\mathcal{J}_{\text{OPO}}(\theta) = \mathbb{E}_{x\sim\mathcal{D}, \{y_i\}_{i=1}^K\sim\pi_\theta(\cdot|x)}\left[\frac{1}{K}\sum_{i=1}^K \log \pi_\theta(y_i|x) \cdot A_i(x, y_i)\right] \quad (11)$$

where the advantage $A_i$ for trajectory $y_i$ is calculated using an empirical estimate of the optimal baseline $b^*(x)$ based on the $K$ samples:

$$A_i = r(x, y_i) - b^*(x)$$
$$b^*(x) = \frac{\sum_{i=1}^K l_{y_i} \cdot r(x, y_i)}{\sum_{i=1}^K l_{y_i}} \quad (12)$$

By normalizing the reward with this optimal baseline, we can minimize the variance of our policy gradient estimates practically, leading to more stable and effective learning. In particular, our objective function omits commonly used KL and entropy regularization terms. We demonstrate that OPO can achieve strong performance even without relying on these regularizations. A detailed summary of the OPO algorithm is provided in Algorithm 1.

---

**Algorithm 1** Optimal on-Policy Optimization (OPO)

---

**Require:** Initial policy model $\pi_{\text{SFT}}$, reward function $r(x, y)$, prompt dataset $\mathcal{D}$
1: policy model $\pi_\theta \leftarrow \pi_{\text{SFT}}$
2: **for** step = 1, 2, ..., N **do**
3:     Sample a batch of prompts $\mathcal{D}_b \sim \mathcal{D}$
4:     For each prompt $x \in \mathcal{D}_b$, sample $K$ responses $\{y_i\}_{i=1}^K \sim \pi_\theta(\cdot|x)$
5:     Compute the advantage $A_i$ for each sampled response $y_i$ using Equation 12
6:     Update the policy model $\pi_\theta$ by maximizing $\mathcal{J}_{\text{OPO}}(\theta)$ defined in Equation 11
7: **end for**
**Ensure:** $\pi_\theta$

---

## 4 EXPERIMENTS

### 4.1 EXPERIMENTAL DETAILS

**Training Setup**   We validate OPO through two sets of comparisons, each designed to isolate the contribution of a key component. The implementation is based on verl[1] (Sheng et al., 2024) training library. To evaluate the impact of exact on-policy training, we compare on-policy GRPO and loose on-policy (off-policy) GRPO training from the *DeepSeek-R1-Distill-Qwen-7B*[2] model. Both variants use a training length of 8k, a learning rate of 1e-6, zero KL penalty, and a batch size of 256 questions. For each question, $K$=16 responses are sampled. For the rollout process, we adopt a temperature of 0.6 and a top-p sampling threshold of 1.0. The mini-batch sizes are 256 (on-policy) and 128 (off-policy). The total training step is 500. We also follow prior work and apply a clip range of 0.2 and a small entropy penalty of 0.001 to the off-policy variant to mitigate entropy collapse, while the on-policy version uses no entropy regularization.

To evaluate the effect of the optimal reward baseline under exact on-policy training, we adopt a more comprehensive and realistic setting. We first perform supervised fine-tuning using long-form chain-of-thought (long-CoT) data, followed by reinforcement learning using both standard on-policy GRPO and our proposed method (OPO), which augments exact on-policy training with the optimal baseline. Both methods share the same hyperparameters: a training length of 24k, a batch size of 256 questions, $K$=8 responses sampled per question, and no KL or entropy terms are applied.

**Training Datasets**   For training datasets, we utilize the math subset from *Skywork-OR1-RL-Data*[3]. This dataset comprises 48k unique math problems, which undergoes an initial offline difficulty estimation for each problem and the problems with all correct or all incorrect responses are excluded. For reinforcement learning, we employ the rule-based reward function (Guo et al., 2025), a reward of 1 for a correct response, and 0 for an incorrect one. The correctness is given by the *Math-Verify* evaluator[4]. For the SFT-then-RL experiments, we exclude duplicates from the OR1 data during the SFT stage, using the remaining 25k samples for RL training.

**Evaluation Setup**   We evaluate model performance on three widely used math reasoning benchmarks: MATH-500, AIME 2024 (MAA, 2024), and AIME 2025. For each dataset, we sampled multiple responses from the model with a maximum response length of 32768, a sampling temperature of 0.6, and a top-p sampling threshold of 1.0. We also use the *Math-Verify* evaluator to assess the correctness. For MATH-500, we sample 8 reasponses for each question, while for AIME 2024 and AIME 2025, we sample 16 responses. The pass@$k$ metric for $k \in \{1, 2, 4, 8, 16\}$ is calculated following the method in Chen et al. (2021).

Beyond accuracy, we also analyze the training dynamics of the entropy of the model's output distribution and the KL divergence between the updated and original models. Given comparable performance, lower KL divergence and higher entropy are preferable. Lower KL divergence indicates

---

[1]https://github.com/volcengine/verl
[2]https://huggingface.co/deepseek-ai/DeepSeek-R1-Distill-Qwen-7B
[3]https://huggingface.co/datasets/Skywork/Skywork-OR1-RL-Data
[4]https://github.com/huggingface/Math-Verify

lower alignment tax (undesirable model changes from alignment), while higher entropy indicates greater sampling diversity.

## 4.2 RESULTS

Table 1: The performance comparision between on-policy and off-policy training.

| Dataset | Method | pass@1 | pass@2 | pass@4 | pass@8 | pass@16 |
|---------|--------|--------|--------|--------|--------|---------|
| **MATH-500** | Off-Policy | 92.97 | 95.60 | 97.14 | **98.20** | - |
| | On-Policy | **93.90** | **96.21** | **97.43** | 98.16 | - |
| **AIME 2024** | Off-Policy | 53.50 | 65.62 | 73.92 | 78.07 | 80.00 |
| | On-Policy | **55.42** | **66.60** | **74.37** | **78.81** | **81.33** |
| **AIME 2025** | Off-Policy | 36.21 | 44.05 | 51.60 | **59.02** | **66.67** |
| | On-Policy | **38.37** | **46.58** | **53.65** | 58.54 | 62.66 |

We first investigate the impact of exact on-policy training. Table 1 presents the average performance over the last five checkpoints (steps 420 to 500, in increments of 20) to reduce evaluation variance. The results show that with the same optimization steps, exact on-policy training consistently improves the pass@1/2/4 scores across all benchmarks, indicating that it yields models with higher precision in generating correct solutions on average. For larger $k$ values, the performance gap between on-policy and off-policy methods narrows, suggesting that while off-policy training can eventually recover correct answers through multiple sampling, but in a less efficient manner. In contrast, on-policy training produces models that are more reliable and require fewer samples to achieve strong accuracy. These findings highlight the importance of aligning the optimization procedure with the exact on-policy distribution, as it not only boosts average accuracy but also reduces the reliance on multiple sampling for robust performance.

Table 2: The performance comparison between OPO and GRPO. Both OPO and GRPO follow the exact on-policy training from the SFT policy.

| Dataset | Method | pass@1 | pass@2 | pass@4 | pass@8 | pass@16 |
|---------|--------|--------|--------|--------|--------|---------|
| | SFT | 94.80 | 96.61 | 97.63 | 98.40 | - |
| **MATH-500** | GRPO | 95.10 | 96.64 | 97.51 | 98.16 | - |
| | OPO | **95.26** | **97.00** | **97.91** | **98.52** | - |
| | SFT | 66.04 | 75.72 | **80.13** | 81.65 | 83.33 |
| **AIME 2024** | GRPO | 67.96 | 75.54 | 79.62 | 81.67 | 83.33 |
| | OPO | **68.50** | **76.10** | 80.06 | **82.12** | **84.00** |
| | SFT | 46.88 | 57.39 | 68.13 | 76.89 | 83.33 |
| **AIME 2025** | GRPO | **50.21** | **61.45** | **70.96** | 77.64 | 81.33 |
| | OPO | 50.00 | 60.88 | 70.37 | **78.02** | **85.33** |

To validate the effectiveness of the optimal reward baseline, we compare the performance of OPO against GRPO and the initial supervised fine-tuned policy. Both OPO and GRPO follow the exact on-policy training and share all hyperparameters. The only difference lies in their advantage estimation: OPO uses the optimal reward baseline as defined in Equation 12, whereas GRPO uses the standard baseline in Equation 2. Table 2 presents the results averaged over the last five checkpoints. As demonstrated, OPO outperforms GRPO across most cases, with its improvements becoming more pronounced at higher $k$ values (e.g., pass@8 and pass@16). On the MATH-500 benchmark, which has 500 test examples and thus much lower evaluation variance, OPO consistently outperforms GRPO across all $k$ in pass@k. For example, OPO achieves 95.26 pass@1 and 98.52 pass@8, surpassing both GRPO and the SFT baseline. The uniform improvements at every $k$ indicate that the optimal reward baseline provides more effective and stable policy updates, leading to consistent gains in a low-variance evaluation setting. In contrast, the AIME 2024 and AIME 2025 benchmarks contain only 30 test examples each, making the results more sensitive to evaluation variance. On these benchmarks, OPO and GRPO show competitive performance at low $k$, with GRPO sometimes

slightly ahead. However, as $k$ increases, OPO exhibits clearer advantages. Notably, OPO achieves the highest pass@8 and pass@16 results on both datasets. These findings highlight that the optimal reward baseline enhances both the stability and generalization of on-policy optimization, making OPO a more reliable and effective approach across diverse evaluation settings.

## 4.3 ANALYSIS

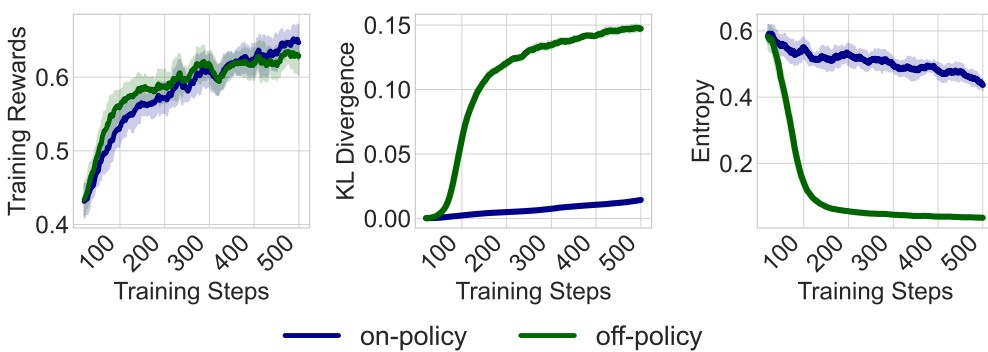

Figure 1: Training dynamics of on-policy and off-policy training. **Left**: Training rewards; **Middle**: KL divergence; **Right**: Entropy.

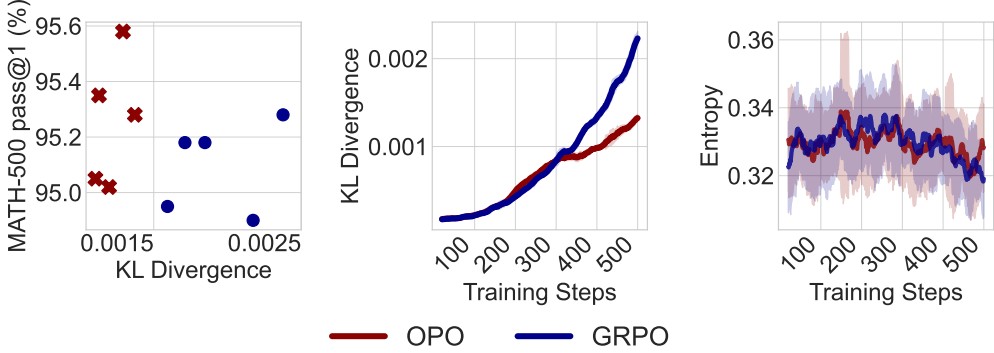

Figure 2: **Left**: Comparison of KL divergence and math performance between OPO and GRPO. Both OPO and GRPO follow the exact on-policy training from the SFT policy. The x-axis represents KL divergence, and the y-axis denotes math performance. **Middle**: Training dynamics of KL divergence. **Right**: Training dynamics of entropy.

**OPO achieves better performance and more stable training.** We compare the training dynamics of on-policy and off-policy methods in Figure 1. While off-policy training achieves similar or even slightly higher training rewards than exact on-policy training in the earlier stage, it yields inferior performance on math reasoning tasks. It suggests a potential overfitting issue with off-policy learning. Furthermore, exact on-policy training exhibits significantly lower KL divergence and higher entropy throughout training, even without any explicit KL or entropy regularization, whereas off-policy training includes an additional entropy bonus. Lower KL divergence implies a reduced alignment tax and higher entropy suggests stronger exploration capability. Figure 2 presents the comparison between OPO and GRPO with exact on-policy training. OPO maintains similar entropy levels while achieving lower KL divergence. The left subplot visualizes the trade-off between KL divergence and math performance, demonstrating that OPO consistently achieves higher performance with more stable training dynamics.

Table 3: Comparison of repetition rate (Rep-5) and sampling diversity (Self-BLEU) between on-policy and off-policy training. Lower values indicate better performance.

| Dataset | MATH-500 | | AIME 2024 | | AIME 2025 | |
|---|---|---|---|---|---|---|
| Method | Rep-5 ↓ | Self-BLEU ↓ | Rep-5 ↓ | Self-BLEU ↓ | Rep-5 ↓ | Self-BLEU ↓ |
| Off-Policy | 18.11 | 74.45 | 25.71 | 69.60 | 27.27 | 69.99 |
| On-Policy | **15.56** | **61.80** | **19.75** | **62.54** | **20.74** | **63.76** |

Table 4: Comparison of repetition rate (Rep-5) and sampling diversity (Self-BLEU) between OPO and GRPO. Lower values indicate better performance.

| Dataset | MATH-500 | | AIME 2024 | | AIME 2025 | |
|---|---|---|---|---|---|---|
| Method | Rep-5 ↓ | Self-BLEU ↓ | Rep-5 ↓ | Self-BLEU ↓ | Rep-5 ↓ | Self-BLEU ↓ |
| GRPO | 14.82 | **66.70** | 22.62 | 64.53 | 22.71 | 63.89 |
| OPO | **14.70** | 66.76 | **22.08** | **64.01** | **21.84** | **63.20** |

**OPO generates more diverse and less repetitive outputs.** The KL divergence and entropy correlate with important output quality metrics that directly impact user experience, such as sampling diversity and repetition rate. We use the Self-BLEU metric (Zhu et al., 2018) to evaluate the diversity of the generated responses. For each query, multiple responses are sampled; each response is treated as a hypothesis and compared to others as references. The average BLEU score across all combinations is reported as Self-BLEU. A lower Self-BLEU score indicates higher diversity among outputs. For measuring the repetition rate of the generated responses, we employ the Rep-5 metric (Welleck et al., 2020), which calculates the proportion of duplicate 5-grams in each generated sequence. A lower Rep-5 score reflects less intra-sequence repetition. Tables 1 and 2 summarize the results. Benefitting from exact on-policy training and the optimal reward baseline, OPO consistently produces outputs that are both more diverse and less repetitive compared to its counterparts.

## 4.4 EXPERIMENTS ON REINFORCE++

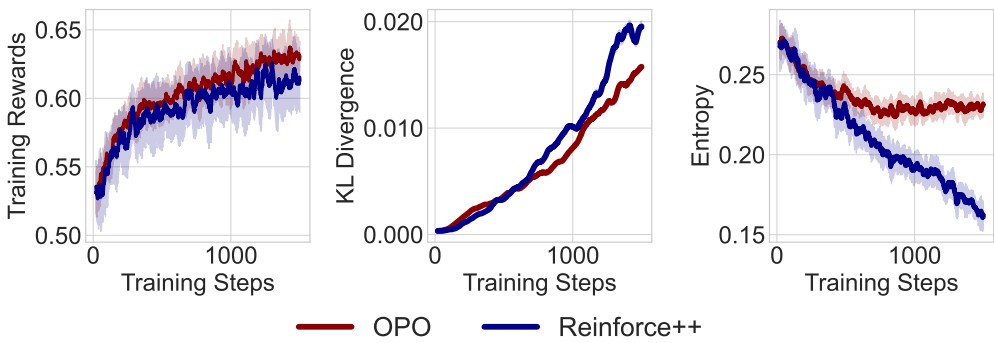

Figure 3: Training dynamics of OPO and Reinforce++. Both OPO and Reinforce++ follow the exact on-policy training. **Left**: Training rewards; **Middle**: KL divergence; **Right**: Entropy.

OPO is a general technique that can be applied to other policy-gradient algorithms. We apply it to the Reinforce++ algorithm (Hu, 2025) to further validate its effectiveness. Unlike GRPO, Reinforce++ utilizes the normalized reward of an entire batch instead of each group as its baseline. We exclude the KL reward in Reinforce++ as exact on-policy training can omit it. For the preliminary experiment, we use Deepseek-R1-Distill-Qwen-1.5B for training, with a response length of 8k and a batch size of 256 questions. We make both OPO and Reinforce++ follow the exact on-policy training. As shown in Figure 3, the training dynamics demonstrate that OPO, by leveraging the length-weighted optimal baseline normalized across each batch, consistently achieves higher train-

ing rewards and maintains higher entropy compared to on-policy Reinforce++. This suggests that the optimal baseline effectively stabilizes training and promotes more diverse policy exploration.

## 5 RELATED WORK

**RL Algorithms** Among various policy-based RL algorithms (Sutton & Barto, 2018), Proximal Policy Optimization (PPO, Schulman et al. 2017) has been the most common choice since Instruct-GPT (Ouyang et al., 2022) due to its balance of stability and sample efficiency. However, PPO needs to train an extra value model to estimate the reward baseline. To address this, Group Relative Policy Optimization (GRPO, Shao et al. 2024) proposes to generate multiple responses and use their average score as a baseline for advantage estimation. It eliminates the need for a separate value model, thereby improving memory efficiency. Other works also focus on alternative advantage estimation methods without a value model, like ReMax (Li et al., 2024), RLOO (Ahmadian et al., 2024), Reinforce++ (Hu, 2025), Dr. GRPO (Liu et al., 2025) and LUFFY (Yan et al., 2025). Furthermore, while some research aims to resolve issues like KL or entropy collapse in loose on-policy settings (He et al., 2025; Yu et al., 2025; Yan et al., 2025), both our method and Chen et al. (2025) emphasize exact on-policy training.

**Variance Reduction in RL** The foundational policy-gradient algorithm REINFORCE (Williams, 1987; 1992; Sutton & Barto, 2018) suffers from high gradient variance. Prior work (Dayan, 1991; Weaver & Tao, 2001; Kakade & Langford, 2002; Greensmith et al., 2004) derive the theoretical optimal baseline that minimizes variance, but the original formulation is impractical in real-world sequence generation scenarios. In the context of LLMs, ReMax (Li et al., 2024) employs a greedy baseline for variance reduction. Other common algorithms apply the mean reward as the baseline. In contrast, we show that under the intuitive assumption, the optimal baseline formulation simplifies to a length-weighted reward, which is feasible for practical use.

**Reinforcement Learning for LLMs** Large Language Models (LLMs) have demonstrated impressive capabilities across a wide range of real-world tasks (Brown et al., 2020; OpenAI, 2023; Anil et al., 2023). A critical phase in their development is Reinforcement Learning from Human Feedback (RLHF, Stiennon et al. 2020; Ouyang et al. 2022; Bai et al. 2022), which typically consists of two stages: supervised fine-tuning (SFT) and reinforcement learning (RL). In SFT, models are initially guided toward preferred behaviors using curated datasets. Subsequently, RL optimizes the model outputs by employing policy gradient algorithms to maximize a reward signal (Gao et al., 2022; Rafailov et al., 2023). It ensures that the model aligns with desired outcomes like helpfulness, truthfulness, and harmlessness. Beyond general alignment, RL has been applied to enhance the reasoning capabilities of LLMs (OpenAI, 2024; Guo et al., 2025; XAI, 2024; DeepMind, 2024). These methods often emphasize test-time scaling, where models iteratively refine their thought processes, explore alternative strategies, and self-correct through chain-of-thought reasoning (Wei et al., 2022). Such techniques significantly boost performance on complex tasks in domains including mathematics, science, and programming.

## 6 CONCLUSION

This paper proposes on-policy reinforcement learning with optimal reward baseline (OPO), which adheres to exact on-policy training and derives the practically feasible optimal baseline for advantage estimation in the basic policy gradient framework. OPO employs a single policy model without relying on KL divergence constraints or entropy regularization, yet achieves superior performance and improved training stability. Furthermore, our results indicate that OPO encourages the generation of more diverse and less repetitive outputs. We have validated the effectiveness of the proposed method on math reasoning tasks using a rule-based reward. For future work, we aim to conduct more extensive experiments across a broader range of reinforcement learning algorithms to assess the generality and robustness of our approach. In addition, we plan to extend the optimal baseline to off-policy reinforcement learning settings to further improve the applicability.

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

## A  DERIVATION OF THE OPTIMAL BASELINE

To reduce the variance of the policy gradient estimate, we consider adding a baseline $b$ to the reward. It does not affect the expectation of the gradient and can reduce its variance. By adding the baseline, the variance of the gradient estimate is given by:

$$\text{Var}[g] = \mathbb{E}\left[(\nabla_\theta \log \pi_\theta(y|x) \cdot (r(x,y) - b))^2\right] - (\mathbb{E}\left[\nabla_\theta \log \pi_\theta(y|x) \cdot (r(x,y) - b)\right])^2$$

Since the second term is independent of $b$, minimizing the variance is equivalent to minimizing the following objective:

$$J(b) = \mathbb{E}_{y \sim \pi_\theta(\cdot|x)}\left[(\nabla_\theta \log \pi_\theta(y|x) \cdot (r(x,y) - b))^2\right]$$

Let us define the shorthand:

$$g(y) := \nabla_\theta \log \pi_\theta(y|x), \quad r := r(x,y)$$

Then the objective becomes:

$$J(b) = \mathbb{E}_{y \sim \pi_\theta(\cdot|x)}\left[(g(y) \cdot (r - b))^2\right]$$

Expanding the square:

$$\begin{aligned}
J(b) &= \mathbb{E}\left[g(y)^2 \cdot (r - b)^2\right] \\
&= \mathbb{E}\left[g(y)^2 \cdot (r^2 - 2rb + b^2)\right] \\
&= \mathbb{E}\left[g(y)^2 r^2\right] - 2b\,\mathbb{E}\left[g(y)^2 r\right] + b^2\,\mathbb{E}\left[g(y)^2\right]
\end{aligned}$$

To minimize $J(b)$, we take the derivative with respect to $b$ and set it to zero:

$$\frac{dJ}{db} = -2\,\mathbb{E}\left[g(y)^2 r\right] + 2b\,\mathbb{E}\left[g(y)^2\right] = 0$$

$$\Rightarrow b^* = \frac{\mathbb{E}\left[g(y)^2 r\right]}{\mathbb{E}\left[g(y)^2\right]}$$

**Conclusion.**  The optimal baseline $b^*$ that minimizes the variance of the policy gradient estimate (for a fixed input $x$) is:

$$b^* = \frac{\mathbb{E}_{y \sim \pi_\theta(\cdot|x)}\left[(\nabla_\theta \log \pi_\theta(y|x))^2 \cdot r(x,y)\right]}{\mathbb{E}_{y \sim \pi_\theta(\cdot|x)}\left[(\nabla_\theta \log \pi_\theta(y|x))^2\right]}$$

This baseline depends on both the policy and the reward and yields the minimum gradient variance.

## B  THE USE OF LARGE LANGUAGE MODELS

Large Language Models (LLMs) were used only as auxiliary tools for improving the presentation of this paper. Specifically, we used them to (i) correct minor typographical errors, (ii) improve grammar and clarity of sentences, and (iii) suggest formatting adjustments for tables and figures to align with standard academic style. No LLMs were used for research ideation, methodological design, analysis, or substantive writing of the paper. The core research contributions, experiments, and the initial draft of the manuscript were conceived, developed, and written entirely by human authors.

