# OpenReview forum: "On-Policy RL with Optimal Reward Baseline"
_ICLR.cc/2026/Conference — ICLR 2026 Conference Withdrawn Submission_

### Official Review · Reviewer_8d7i · 2025-10-30

**Soundness:** 2
**Presentation:** 2
**Contribution:** 1
**Rating:** 2
**Confidence:** 3

**Summary:**

This paper proposes On-Policy RL with Optimal reward baseline (OPO), a reinforcement learning algorithm. OPO aims to address training instability and computational inefficiency encountered in current methods for aligning large language models (LLMs) and enhancing LLM reasoning capabilities. The algorithm incorporates two main features: exact on-policy training and a practical formulation of an optimal reward baseline. The optimal reward baseline minimizes policy gradient variance. Experimental results on mathematical reasoning benchmarks indicate that OPO has improved performance and training stability. OPO operates without requiring auxiliary models or explicit regularization terms. The algorithm shows lower policy shift and higher output entropy, resulting in more diverse and less repetitive generated responses.

**Strengths:**

1. The algorithm saves the need for auxiliary models (like value networks) and explicit regularization terms (such as KL divergence or entropy bonuses), simplifying the training pipeline.

2. Experimental results show that OPO achieves improved performance on mathematical reasoning benchmarks and maintains stable training dynamics with lower policy shift and higher output entropy.

**Weaknesses:**

1. The formula for the optimal baseline, which weights the reward by the squared magnitude of the score function, is a very well-known result in variance reduction for policy gradient methods (REINFORCE) [1]. This considerably undermines the originality of this work.

2. Around Eq. 10, the authors make two strong and unjustified assumptions: one is the orthogonal token gradients, and the other is the same distribution of different squared token gradient norms. The first assumption hardly holds in the context of sequence models (LLMs) because:

a) Shared Parameters: All tokens in the sequence are generated by the exact same set of parameters ($\theta$). The gradient $\mathbf{g}_t$ is the effect of token $y_t$ on all parameters. Since a single word (like ``not'') might significantly influence the probability distribution for many subsequent words, the gradients are highly correlated across time, meaning that they are far from orthogonal.

b) Context Dependence (Autoregressive Structure): The gradients $\mathbf{g}_t$ and $\mathbf{g}_k$ are dependent because

$\\pi_\\theta(y_k|y_{1:k-1}, x)$ depends on the entire history, which itself depends on the parameters. In an autoregressive model, the policy for generating $y_k$ relies on the embeddings generated by $y_{1:k-1}$, making the gradients intrinsically linked.

c) Covariance Matrix is Non-Diagonal: If the gradients were truly orthogonal, the expected Fisher Information Matrix (which is related to $E[(\\sum \mathbf{g}_t)(\\sum \mathbf{g}_k)^T]$) would be a diagonal matrix, but in reality, it is highly structured and full of covariance terms.

The second assumption is also unreliable. The gradient magnitude $\\|\mathbf{g}_t\\|^2$ is inversely related to the policy's confidence in its prediction $y_t$. In brief, high confidence corresponds to a small gradient magnitude, while low confidence corresponds to a large gradient magnitude.


3. Mathematical expression incorrectness: the ``squared gradients $(\cdot)^2$'' in Eqs. 8 and 9 seem to be corrected as squared l_2 norm $\\|\cdot\\|^2$.

[1] Barto, A. G. (2021). Reinforcement learning: An introduction. by richard's sutton. SIAM Rev, 6(2), 423.

Based on these significant problems, this work does not meet the quality standard of ICLR yet.

**Questions:**

See weaknesses.

---

### Official Review · Reviewer_5pnU · 2025-10-30

**Soundness:** 2
**Presentation:** 3
**Contribution:** 2
**Rating:** 4
**Confidence:** 4

**Summary:**

The paper proposes "On-Policy RL with Optimal Reward Baseline," highlighting the importance of on-policy reinforcement learning algorithms while introducing a novel reward term to minimize gradient variance. Ultimately, OPO achieves smaller policy shifts and higher output entropy.

**Strengths:**

1. The design of the reward based on minimizing gradient variance is highly innovative.
2. OPO achieves relatively stable training curves without requiring KL penalty or entropy regularization.

**Weaknesses:**

1. The theoretical assumptions in the paper—that "the gradients of different tokens are approximately orthogonal" and "the norm of the gradient for each token follows the same distribution"—lack justification. There is neither empirical statistical validation nor theoretical support, casting doubt on their reliability.
2. A significant portion of current RL algorithms are already based on on-policy methods. If the authors aim to emphasize the importance of on-policy learning, it would be more appropriate to compare with inherently off-policy algorithms like DPO. Additionally, while the authors critique PPO for not being entirely on-policy, a comparison with PPO remains valid.

**Questions:**

1. I believe that an increase in KL divergence does not necessarily indicate training instability, as deviation from the initial distribution is inevitable during training. Therefore, OPO's slower rate of KL increase may only suggest more conservative optimization. How do the authors view this point? I am also curious about the effects of directly using KL penalty or entropy penalty.
2. I observed that even after RL training, the performance improvement is not significant. What might be the reason for this? Are there results with more training steps?

---

### Official Review · Reviewer_ivp9 · 2025-10-31

**Soundness:** 1
**Presentation:** 3
**Contribution:** 1
**Rating:** 0
**Confidence:** 4

**Summary:**

The authors propose OPO, which is a combination of GRPO with an added baseline to reduce variance.

**Strengths:**

I found the paper to be lacking in technical novelty. It is essentially a rebranding of various previous recipes in a new form.

However, I found the interest in Variance Reduction techniques in LLM finetuning to be quite refreshing.

**Weaknesses:**

A Substantial Literature review has been established on variance reduction techniques, though they are good for theoretical results; incorporating baselines rarely changes anything. This is kind of reflected in the training dynamics, as shown in Figure 1. Honestly, I am a bit surprised to see that both on-policy and off-policy training have similar error plots, but I guess that's OK. I might be biased in my opinion, but I think that variance reduction techniques bring little value in RL, and this paper has failed to convince me otherwise.

I think the work would be better suited as a theory paper, see
1. Peter L Bartlett and Jonathan Baxter, Estimation and approximation bounds for gradient-based reinforcement learning,
2. Evan Greensmith, Peter L Bartlett, and Jonathan Baxter, Variance reduction techniques for gradient estimates in reinforcement learning,
3. P. Marbach and J. N. Tsitsiklis, Simulation-based optimization of markov reward processes,
4. P. Marbach and J. N. Tsitsiklis, Simulation-based optimization of markov reward processes,
5. Jonathan Baxter and Peter L Bartlett, Infinite-horizon policy-gradient estimation,
6. Peter L Bartlett and Jonathan Baxter, Estimation and approximation bounds for gradient-based reinforcement learning,

**Questions:**

Q1. Can you explain how you derived equation 10 from the assumptions?
Q2. Any particular reason you are including $\log \pi$ in the OPO objective? There seems to be a lack of transparancy between the text before which includs GRPO and variance reduction to the formulation.

---

### Official Review · Reviewer_Akfd · 2025-11-01

**Soundness:** 3
**Presentation:** 3
**Contribution:** 2
**Rating:** 4
**Confidence:** 3

**Summary:**

This paper proposes On-Policy Reinforcement Learning with Optimal reward baseline (OPO), a novel and simplified reinforcement learning algorithm. There are two key innovations: 1) exact on-policy training to ensure that every policy update is computed with freshly sampled trajectories to maintain stability and prevent entropy collapse; and 2) a practical approximation of the optimal reward baseline that minimizes gradient variance by weighting rewards by sequence length. The authors evaluate OPO on mathematical reasoning benchmarks (e.g., MATH-500, AIME 2024, AIME 2025) and show that OPO achieves higher accuracy, lower policy shifts as indicated by reduced KL, and higher output entropy, resulting in more diverse, less repetitive generations. These results show the potential of OPO as a promising direction for stable and effective reinforcement learning in large language model alignment and reasoning tasks.

**Strengths:**

1. The paper is well motivated, addressing an important research problem, RL algorithms often suffering from training instability due to loose on-policy constraints.
2. The paper is well written, the derivation of the proposed OPO algorithm is easy to follow and understand.
3. Optimal reward baseline is simple and straightforward to implement, making it easy to incorporate into various RL algorithms.
4. The empirical results seems convincing: experiments on math benchmarks show that OPO achieves higher accuracy, reduced KL, and greater output entropy

**Weaknesses:**

1. The evaluation is limited to math benchmarks only; it would be nice to include results on code, science, STEM, or instruction-following benchmarks as well.
2. A key assumption in the derivation of OPO is that the gradients of different tokens are approximately orthogonal and that the gradient norms follow the same distribution. There is a lack of explanation and evidence for why this assumption would hold.
3. While the authors demonstrate the improvement brought by the optimal reward baseline when integrated into GRPO, there is a lack of comparison and discussion on whether it would be compatible with other RL algorithms.
4. While the authors demonstrate better performance with exact on-policy training, this approach can be computationally expensive compared to off-policy variants that reuse trajectories. The paper lacks discussion and ablation on compute cost.
5. The paper lacks comparisons against other strong RL baselines such as PPO, DPO, and Dr. GRPO.

**Questions:**

Can you illustrate more on the assumption in the derivation of OPO that "the gradients of different tokens are approximately orthogonal and the norm of the gradient for each token follows a same distribution"?

---

### Note · Authors · 2025-12-03

I have read and agree with the venue's withdrawal policy on behalf of myself and my co-authors.